# Learning to Cycle: Why Is the Balance Bike More Efficient than the Bicycle with Training Wheels? The Lyapunov’s Answer

**DOI:** 10.3390/jfmk9040266

**Published:** 2024-12-10

**Authors:** Cristiana Mercê, Keith Davids, Rita Cordovil, David Catela, Marco Branco

**Affiliations:** 1Sport Sciences School of Rio Maior, Santarém Polytechnic University, Avenue Dr. Mário Soares No. 110, 2040-413 Rio Maior, Portugal; catela@esdrm.ipsantarem.pt (D.C.); marcobranco@esdrm.ipsantarem.pt (M.B.); 2Physical Activity and Health—Life Quality Research Centre (CIEQV), Polytechnique University of Santarém, Complex Andaluz, Apart 279, 2001-904 Santarém, Portugal; 3Sport Physical Activity and Health Research & Innovation Center (SPRINT), Santarém Polytechnic University, Complex Andaluz, Apart 279, 2001-904 Santarém, Portugal; 4Interdisciplinary Center for the Study of Human Performance (CIPER), Faculty of Human Kinetics, University of Lisbon, Cruz Quebrada-Dafundo, 1499-002 Lisboa, Portugal; cordovil.rita@gmail.com; 5Sport & Human Performance Group, Sheffield Hallam University, Sheffield S10 2BP, UK; k.davids@shu.ac.uk; 6Quality Education—Life Quality Research Centre (CIEQV), Santarém Polytechnique University, Complex Andaluz, Apart 279, 2001-904 Santarém, Portugal

**Keywords:** bicycle, functional variability, nonlinear, inertial sensors, skill adaptation, postural control, learning paths, motor development, physical activity, health

## Abstract

Background/Objectives: Riding a bicycle is a foundational movement skill that can be acquired at an early age. The most common training bicycle has lateral training wheels (BTW). However, the balance bike (BB) has consistently been regarded as more efficient, as children require less time on this bike to successfully transition to a traditional bike (TB). The reasons for this greater efficiency remain unclear, but it is hypothesized that it is due to the immediate balancing requirements for learners. This study aimed to investigate the reasons why the BB is more efficient than the BTW for learning to cycle on a TB. Methods: We compared the variability of the child–bicycle system throughout the learning process with these two types of training bicycles and after transitioning to the TB. Data were collected during the Learning to Cycle Program, with 23 children (6.00 ± 1.2 years old) included. Participants were divided into two experimental training groups, BB (N = 12) and BTW (N = 11). The angular velocity data of the child–bicycle system were collected by four inertial measurement sensors (IMUs), located on the child’s vertex and T2 and the bicycle frame and handlebar, in three time phases: (i) before practice sessions, (ii) immediately after practice sessions, and (iii), two months after practice sessions with the TB. The largest Lyapunov exponents were calculated to assess movement variability. Conclusions: Results supported the hypothesis that the BB affords greater functional variability during practice sessions compared to the BTW, affording more functionally adaptive responses in the learning transition to using a TB.

## 1. Introduction

Riding a bicycle has recently been considered a foundational movement skill [1,2] with multiple lifetime benefits [3], which should be promoted early in development [4]. The new conceptual model proposed by Hulteen et al. [1] suggests replacing the term ‘fundamental movement skills’ with ‘foundational movement skills’, which underpins a significant conceptual adaptation to broaden the scope of skills, including specifically riding a bicycle, considered important for promoting physical activity and other positive health trajectories across a lifespan. Indeed, several studies corroborate that learning to cycle supports adherence to and the maintenance of healthy and positive trajectories, particularly due to the beneficial effects on physical health, such as improved body composition and enhanced cardiorespiratory fitness [3], and mental health, including increased socialization opportunities and the development of social skills [5,6]. For all these reasons, previous studies have considered learning to ride a bicycle an important milestone [4,7,8].

According to research, even though using a bicycle with lateral training wheels (BTW) is the most common approach to learning to cycle worldwide [9], the balance bike (BB), a bicycle without pedals nor training wheels, is the most efficient learning bicycle [4,9,10,11,12,13]. It has been argued that the use of the BTW is a mistake [12] which can be counterproductive [10,11,14,15,16,17,18,19]. By adding the side wheels, as an artificial method to increase stability and minimize bicycle oscillations, the child learns to pedal without directly experiencing the bicycle’s instability. When training wheels are removed, the child is confronted for the first time with the instability of the bicycle and responds by organizing defensive responses, including freezing their upper limbs and trunk, impacting the imbalance of the bicycle [10,13]. On the other hand, a child using the BB from the start immediately learns to engage with instability and only has to deal with pedaling after achieving and maintaining balance. Several other studies have indicated that the balance bike (BB) is a more efficient learning tool than the bicycle with training wheels (BTW). For instance, a recent systematic review investigated the best strategies to promote efficient cycle learning [4]. Additionally, a retrospective study found that children who learn to cycle using the BB can successfully cycle independently on average 1.81 years earlier than those who use the BTW [9]. Despite the significance of these ideas [4,12,20] for learning to cycle only, one study has sought to experimentally verify the efficacy of these two training bicycles [13]. The investigation applied an intervention program (L2Cycle), comparing two groups of kindergarten and elementary school children, where one group practiced with the BTW and another with the BB. They found that the BB group learned to self-start, ride, brake, and cycle independently (i.e., all these cycling milestones performed sequentially) significantly faster than participants in the BTW group, which corroborates previous suggestions in the literature. However, the “why” question is left unresolved, with a need for research to verify possible reasons behind the greater effectiveness of the BB in learning to cycle.

Our conceptualization of the BB’s learning effectiveness lies in the functionality of motor system variability for skill adaptation. Traditional approaches to interpreting movement system variability considered it ‘noise’ or a result of errors [21,22] to be eliminated from performance. More recently, a dynamical systems interpretation has highlighted the functionality and importance of movement variability for adapting skills [23]. That is, the same coordination task, like cycling with a training bicycle, could be performed by re-organizing multiple elements or degrees of freedom (e.g., motor units, muscles, joints, limbs, a movement axes, and planes) and a wide variety of combinations between them [24]. Functionality is assumed to be a system’s ability to carry out its tasks effectively and adaptively. Movement variability, as a movement system that explores different solutions for the same task, can contribute to task functionality by affording system adaptability in facing unexpected and challenging situations [25]. The human movement system has evolved the capacity to produce several solutions for the same coordination task (e.g., locomotion), affording functional system adaptability in facing unexpected and challenging contexts, such as being able to use a traditional bicycle [21,25]. Variability during the learning process is currently considered a crucial aspect and is one of the essential elements of the recent theory of nonlinear pedagogy [23,26]. According to this theoretical framework, which is based on dynamic systems [27,28], Newell’s constraints [29], and ecological dynamics [30], learning should be learner-centered rather than teacher-centered. The teacher/coach acts as a filmmaker who sets the stage for learning (i.e., manipulates various constraints) so that the main actor, the learner, can self-organize and acquire and master the new motor skill. Therefore, the teacher should introduce variability during the process, encouraging the learner to seek new and more efficient motor solutions [23,26].

Movement variability has traditionally been measured by using linear tools, like the standard deviation statistic, to quantify the amount of variability independently of their order in a data series [22]. In contrast, nonlinear methods afford analysis based on the performance process, looking for both the structure and quality of variability [31]. To analyze variability in biological systems, as in the case of a child riding a bicycle, nonlinear measurement tools can provide deeper insights in the (re)organization of movement [31]. In this sense, several nonlinear methods could be used, such as recurrence quantification analysis (RQA) [32], which evaluates the recurrence of dynamic states in time series; or, the single scale entropy could be used, which can be used as a measure of the uncertainty and irregularity of time series [33,34]. Nevertheless, considering the present study’s purpose, which consists of specifically analyzing motor variability, and its specifications, including periodic data from angular velocities, the most suitable nonlinear technique consists of the largest Lyapunov exponent (LyE) [35,36]. The LyE is probably the most popular nonlinear method used to assess stability and variability [31,35]. This method is widely used in the analysis of biological systems because it offers a deeper understanding of the neuromotor control of movement. It is very sensitive to initial conditions and the divergence of trajectories in dynamic systems, providing a robust measure of system stability and variability [31,35,36]. This method reconstructs the data in a system’s state phase and measures the rate at which the orbits converge or diverge. In periodic signals, the LyE value is 0, as the orbits do not converge or diverge. A positive LyE signifies that the orbits are diverging, while a negative value indicates that the orbits are converging [37]. The LyE has already proven to be a valid measure for analyzing human gait [35,36], since lower values LyE indicate rigidity in the system and an inability to adapt. In contrast, higher values indicate greater variability and adaptability, with the system being able to respond more quickly to destabilization in order to maintain system order [35,38].

Considering the gap in the literature regarding rationalizations for the greater efficiency of BBs, compared to BTWs, and the potential of using nonlinear methods to study rider–system variability, the present study sought to investigate the process of learning to cycle with either the BB or BTW. Specifically, we compared variability (by using the LyE) (i) within the same training bicycle group (BB or BTW), (ii) between bicycle groups (BB vs. BTW) at different stages of learning, and (iii), between children who did and did not learn to cycle independently. We also considered the newer theories that have highlighted the importance of variability [39] and the existing literature that points out balance exploration as the key component for the BB’s efficiency [4,40]. We hypothesized that the mechanisms behind the effectiveness of the balance bike (BB) include the immediate engagement with balance and postural control, which promotes greater functional variability. This variability may allow children to explore and adapt their movements more effectively, leading to the quicker mastery of cycling skills. In this sense, we hypothesized the following: (a) the BB would afford greater functional variability, compared to the BTW, during the period of first contact with a bicycle and after training; (b) there would be no difference in variability after children learned to independently cycle on the traditional bicycle; and (c) children who did not learn to cycle independently during the program would display lower values of variability than children who did. If these hypotheses were confirmed, the findings could indicate that the greater values of variability in using the BB signal its greater efficiency in supporting children’s skill adaptation in functionally using a traditional bicycle.

## 2. Materials and Methods

### 2.1. Study Design

This study was conducted during the Learning to Cycle Program (L2Cycle) intervention, a two-week bicycle camp that helped children to learn how to cycle. The program included six lessons with a BB group and a BTW group, followed by four sessions with both groups using the traditional bicycle (TB) (i.e., with pedals and without training wheels). The training environment was designed with different surfaces, slopes, and vertical obstacles, allowing children to explore actions related to the basic milestones of cycling including self-starting, maintaining balance, moving around, avoiding obstacles, and braking. Practice sessions were undertaken daily for a duration of 30 min and were conducted by physical exercise technicians with safety equipment [13]. The results regarding the effectiveness of learning to cycle between the BB and BTW, by analyzing the number of sessions required to achieve each stage of cycling, as well as independent cycling, have already been presented and discussed in a previous study (see more information in [13]). This intervention was simultaneously used to collect data on variability during the learning process using two different training bikes, which is the focus of the present study.

To analyze movements during the learning process, three evaluation phases were defined: (i) before the training program, with the training bicycle (O1), (ii) after the six training lessons, still with the training bicycle (O2), and (iii), two months after the training program, with the TB; see Figure 1. The program and the data collection procedures were approved by the Ethics Committee of the Faculty of Human Kinetics (approval number: 22/2019).

### 2.2. Participants

Twenty-three children participated in the study, aged between three and seven years (nine girls; M= 6.00; SD = 1.20 years). Random stratified samples were constituted based on sex and age. Twelve children were allocated to the BB group and eleven to the BTW group. In O1 (pre-intervention evaluation), no statistically significant differences were found between the groups regarding their weight, height, BMI, BMIs percentile, motor competence, and previous bicycle experiences (all *p_s_* > 0.05) [13]. Participation in the study was completely voluntary and free of charge, and participants could leave the study if they wished without having to give reasons.

Before the intervention, we confirmed that none of the participants were able to cycle independently, meaning that they did not know how to ride a traditional bicycle. The ability to cycle independently prior to the intervention was an exclusion criterion. To be considered independent riders, the children needed to fulfill the following criteria: to be able to perform a self-launch (to start cycling, the researcher could only stabilize the bicycle if the child’s feet could not reach the ground, due to small stature), cycle for at least 10 m, and brake safely.

All participants performed in O1 and O2. However, in O3, four children in the BB training group did not participate, with one due to health issues and three because they were not able to perform the self-launch, due to their small stature. Another four children in the BTW group did not participate in O3, with one because of their small stature and three because they could not cycle independently using the TB.

### 2.3. Bicycle Equipment

The bicycles used were the LittleBig Balance Bike (LittleBig, Wicklow, Ireland). This model was chosen because it can be adapted, through the rotation of its saddle, to children from 2 to 7 years old and because it allows the insertion of the pedal crank in it. The BTW group used the same LittleBig model but with the pedal crank and two lateral training wheels added. The use of the same bicycle model in the two groups also allowed us to eliminate possible variables masked by use of different bicycle models by the different groups, ruling out variations due to ergonomic issues or component friction.

### 2.4. Data Collection and Protocols

To ensure similarity between the groups at the beginning of the intervention, data on body composition, motor competence, and previous cycling experience were collected during the O1 evaluation. A Level II anthropometrist certified by the International Society for the Advancement of Kinanthropometry (ISAK) performed the body composition measurements, which included weight and height [41], followed by the calculation of BMI and its percentile classification according to age and sex [42]. Motor competence was assessed using the Motor Competence Assessment battery [43]. Previous cycling experience information was gathered through a brief questionnaire administered to parents or guardians [13].

To ensure that the participants were comfortable and familiar with the study procedures and equipment, a thorough familiarization process was conducted prior to the data collection [44]. This process included the following steps: (i) an introduction to the task, presented as a game; (ii) equipment familiarization, where the participants were introduced to the customized vest with the inertial measurement units (IMUs) and the equipment was presented as a superhero outfit that was going to collect some information about the game they were playing; (iii) equipment exploration, where, prior to mounting the bicycle, the children were invited to walk and run a little with the customized vest in order to get used to the feel of the equipment; (iv) safety and comfort checks, where, throughout the familiarization process, investigators conducted regular checks to ensure that the equipment was properly fitted and that the participants were comfortable.

During all moments of evaluation (O1, O2, and O3), the children were invited to ride a bicycle for five minutes, in a 10 m × 10 m area, with no further instructions or feedback from trainers (for more details, see [45]).

Considering that we wanted to calculate the variability of the child and the bicycle, four inertial measurement units (IMUs) (SparkFun 9DoF Razor, Niwot, CO, USA) were used and placed in specific locations. These specific locations were chosen to provide a comprehensive analysis of both the rider’s body movements and the bicycle’s dynamics, allowing for a detailed understanding of the child–bicycle system during the learning process. On the participants, one IMU was placed at the vertex point through an adjusted headband [46,47], allowing the analysis of head movements. This location was selected because head movements are crucial for understanding how the child controls their balance and spatial orientation while cycling. Another IMU was placed by the second vertebra of the thoracic column (T2) [48], through a customized vest, allowing analyses of trunk segment motions. The trunk plays a fundamental role in maintaining balance and coordinating movements during cycling. By monitoring the trunk’s movements, we could assess how the child stabilized their body and adapted their posture to maintain equilibrium and control the bicycle effectively. On the bicycle, one IMU was placed in the spokes of the front wheel, providing data from the handlebar. This location was chosen because the handlebar movements were directly influenced by the child’s actions and were essential for steering and maintaining the bicycle’s stability. Analyzing the handlebar data helped us understand how the child manipulated the bicycle to navigate and maintain balance, which is critical for developing cycling proficiency. Another IMU was placed in the seat tube of the bicycle frame, providing motion data relating to the whole bicycle [11]. This location was selected to capture the overall dynamics of the bicycle, including its oscillations and adjustments made by the child; see Figure 2. Inertial sensors (IMUs) have been consistently shown in the previous literature to be reliable tools for capturing kinematic data [49]. These small, powerful devices allow for the quick and convenient collection of large amounts of data, enabling more ecological assessments. Winter et al. [50] highlighted their effectiveness in analyzing movement variability during cycling, demonstrating their capability in capturing kinematic data across different body segments. The excellent validity and reliability of IMUs have also been tested and proven in various other activities, such as running and walking [51,52] and jumping [53]. The sensor was calibrated according to the manufacturer’s recommendations by holding the sensor in a static position for 8 s previous to each collection [54]. All collected data from the IMUs were sampled at a rate of 100 Hz, with a full scale of 4G defined for the accelerometer and 2000 deg/s for the gyroscope [45].

Each data collection phase was also video-recorded to identify the points in time when the child was cycling or performing other activities (e.g., stopping to rest or fall). The video was recorded with a smartphone (Samsung A71, Seoul, Republic of Korea) at 30 Hz. To synchronize the IMUs and the video, before each data collection period, the researcher lifted and dropped the bicycle’s front wheel on the ground, on three consecutive occasions, with intervals of approximately five seconds between them. Before data analysis, a visual inspection was made to identify the three peaks corresponding to the three drops. The IMUs were synchronized by identifying the first acceleration peak from each drop. The video was also synchronized to the IMU data by identifying the first video frame corresponding to the first impact of the front wheel on the ground; this synchronization process allowed us to identify in the IMU data the onsets of periods of cycling and other activities previously verified in the videos [45].

### 2.5. Data and Statistical Treatment

Initially, all video clips were analyzed to identify the beginning and end of each data collection episode, as well as the moments when the child was not cycling (e.g., when they fell or their image exited the video frame). These were later discarded.

Data treatment was performed with a custom matlab routine. Considering that the length of a time series affects the calculation of LyE values, and following the recommendation that time-normalization to a fixed point or duration is necessary [36], all time series were cut according to the one with the lowest duration, which was fixed at three minutes. This value is within methodological recommendations established in the literature [36]. After normalization, the data were filtered using a low-pass, second-order Butterworth filter with a cutoff frequency of 10 Hz (e.g., [55,56]). Therefore, the LyE values for the angular velocity were calculated for each IMU, child, and evaluation moment. The angular velocity variable was chosen because it allows the study of postural control (e.g., [57,58]) and because the movements under analysis were mainly rotations.

For the statistical analysis of the data from the IMUs located on the children’s heads and trunks, three movement planes were considered, since the movement was multiplanar. For the handlebar IMU, only the frontal and transverse planes were considered, since the handlebars did not move in the sagittal plane. For the IMU of the bicycle frame, only the frontal plane was considered since it did not move in sagittal and transversal planes.

The statistical analysis of the data was performed with the Statistical Package for Social Sciences (IBM Corp, version 26), and the statistical significance level was defined at *p* < 0.05. Descriptive statistics were used for sample characterization and for the LyE values of each IMU for each movement plane, evaluation moment, and group. The Shapiro–Wilk test was used to estimate the samples’ normality of data distribution. Accordingly, paired *t*-tests were used to compare, within the same group, the values of each IMU, for each movement plane, between the different evaluation moments. Independent *t*-tests were performed to compare the same IMU, for each movement and evaluation moment, between the two training groups, BB and BTW, and between the BTW children who became independent riders and those who did not. For all significant comparisons, the effect size *r* was calculated [59].

## 3. Results

The descriptive data (average ± standard deviation) of each IMU for each movement plane, moment of evaluation (O1, O2, and O3), and group are presented in Table 1. All Lyapunov mean values were highly positive and far from zero, signifying divergent orbits or that the body and bicycle oscillations were not regular in space. These values were always higher in the BB group, except for frontal plane in the O1 and O2 moments. Per group and between movement planes, the Lyapunov standard deviations of the child and of the bicycle were small and similar, particularly very much smaller than the Lyapunov means. This finding is statistically important, considering the sample sizes. Interestingly, when the standard deviation values were analyzed, the BTW group presented higher values than the BB group for both O1 and O2, except only at the O2 point’s sagittal plane. However, in O3, this tendency was inverted, and the BB group had higher standard deviation values, with the only exception of the vertex’s frontal plane.

### 3.1. Comparisons Between Evaluation Moments

Considering the BB group, between pre-intervention (O1) and after six training sessions (O2), there were significant increases in the variability in the following: (i) the bicycle frame for lateral rotations in the frontal plane (t(11) = −2.41; *p* = 0.035; r = 0.588); (ii) the handlebar for lateral rotations in the frontal plane (t(11) = −3.74; *p* = 0.003; r = 0.748); and (iii) left–right rotations in the transverse plane (t(11) = −2.334; *p* = 0.04, r = 0.576). No statistically significant changes were observed for the data from the vertex and T2 IMUs. In comparing results from after training with the BB (O2) to results from after cycling skill acquisition on the TB (O3), significant decreases occurred in the variability levels at T2 for all planes; the children reduced their velocities for flexion and extension in the sagittal plane (t(7) = 2.634; *p* = 0.034; r = 0.706), lateral flexions in the frontal plane (t(7) = 4.201; *p* = 0.004; r = 0.775), and left–right rotations in the transverse plane (t(7) = 2.467; *p* = 0.043, r = 0.682). No significant changes were verified for the other IMUs. The statistical outcomes are presented above in Table 2, and an illustrative schematic of all comparisons is presented in Figure 3.

Considering the BTW group, after the six training sessions, there were significant increases in the system variability in the following: (i) the vertex for left–right rotations (t(10) = −2.636; *p* = 0.025; r = 0.640); (ii) the handlebar for lateral rotations (t(10) = −3.218; *p* = 0.009; r = 0.713); and (iii) left–right rotations (t(10) = −3.077; *p* = 0.012; r = 0.697). There were no significant changes observed in the data from the T2 and bicycle frame IMUs. In comparing results from after training with the BB (O2) to results from after cycling skill acquisition on the TB (O3), significant increases in the variability for the T2 IMU were verified in the values for the velocity of the flexion and extension (t(6) = −3.152; *p* = 0.020; r = 0.790) and in the bicycle frame’s velocity of lateral rotations (t(6) = 4.219; *p* = 0.006; r = 0.865); no differences were observed in the variability values for the vertex nor the handlebar (see Figure 3).

### 3.2. Comparisons Between Groups

The comparison between the BB and BTW groups showed significantly greater variability in the BB group during O1 and O2. This was observed for the vertex (flexion and extension, lateral oscillations, and left–right rotations), T2 (flexion and extension, lateral oscillations, and left–right rotations), and handlebar (lateral oscillations and left–right rotations). Conversely, the BTW group showed greater variability in the bicycle frame’s lateral oscillations during both O1 and O2, confirming the Lyapunov mean values. The statistical results for significant differences are presented in Table 3 and Figure 3.

In O3, after acquiring cycling skills, the only difference between the groups was in the velocity of the bicycle frame’s lateral oscillations, with the BB group showing greater variability.

### 3.3. Comparisons to Children That Did Not Acquire the Skill of Cycling Independently

The L2Cycle program demonstrated an 88% success rate for achieving independent cycling on the TB. Notably, all children in the BB group successfully reached the level of independent cycling, resulting in a 100% success rate. In contrast, three children in the BTW group did not reach this level, resulting in a 75% success rate. To consider whether variability could be one of the reasons for this failure, a comparison between the performance of children in the BTW group who became independent riders and those who did not was undertaken at the first moment of evaluation, O1, and after six training lessons, in O2. In O1, there was no significant difference in the variability measures recorded by any IMU. At the beginning of training (O1), the BTW children who ended up not achieving independent cycling acted similarly to those who did. However, in O2, the independent riders had a higher level of variability in the velocity of the handlebar for lateral oscillations (t(9) = 4.411; *p* = 0.002; r = 827) and left–right rotations (t(9) = 4.191; *p* = 0.002; r = 0.813); no other statistically significant differences were observed in O2. It should be noted that children who did not acquire the level of cycling independently displayed lower mean values than the independent riders for all other measures from the IMUs and planes of movement; see Table 4.

## 4. Discussion

Despite the widespread use of the BTW, several previous studies suggest that the BB may be more effective in facilitating the transition to the TB. However, the reasons behind this greater effectiveness remain unclear, with the assumption that it may be linked to their inherent exploration of balance. In the present study, we sought to analyze the movement variability that emerged from participants during the process of learning to cycle, at different evaluation times, using the BB or the BTW training cycles, as well as after the cycling skill acquisition with the traditional bicycle (TB). We hypothesized that the BB would provide greater functional variability, which, in turn, could favor skill adaptation and more effective learning outcomes. The greatest exponent of LyE, as a nonlinear measure of movement variability, did emerge as a sensitive system parameter, increasing in value after six training sessions in both groups, recorded at several points on the body and the bicycle, with differing movement planes. Since the LyE measure was calculated through the variable of the angular velocity, its increased variability implied greater and faster variations in rotations, which could represent the bicycle’s and children’s postural oscillations. The implication is that, after training, the children were adapting their postural regulation and exploring more (and faster). This performance characteristic was reflected in increases in the velocity of the left–right rotations of the head on the BTW. In controlling the bicycle, movement variability as a skill adaptation was reflected in increases in the velocities of the handlebar’s lateral oscillations and left–right rotations in both groups and in the velocity of the bicycle frame’s lateral oscillations in the BB group. These increments in movement variability measures could reflect the children’s search of, exploration of, and adaptation to a more freely movable instrument for locomotion, the bicycle. Also, it is interesting to note that not only were patterns of increased movement variability common to both groups but also were their variations. While between O1 and O2 the BB children increased their exploration of the bicycle frame’s control, the BTW children did not. The BB group also displayed increases in the variability of head segment rotations. The use of the BTW when learning to cycle presents similarities with the use of baby walkers in the process of learning to walk. Infants/children sit on the walker/bicycle and just need to walk/pedal without having to worry about balance control or lateral oscillations. Although the use of baby walkers is still not consensual [60], some studies have argued that they delay the child’s movement development with respect to walking [61,62]. Other studies have not confirmed a developmental delay in using walkers but have instead identified kinematic changes in gait patterns [63]. It could be argued that artificial support for movement systems’ postural regulation and stability during locomotion and transport, which both aids provide, when walking or cycling may not provide the necessary task constraints for a child to self-organize and adapt their movement skills for achieving a new task or locomotion pattern.

When analyzing different stages of the learning process, comparing the end of the practice period with the learning bicycles (i.e., BB or BTW, O2) to the end of the period of practice with the TB (O3), it was observed that the variability pattern differed among the two groups; while the BB children reduced their levels of variability in the trunk velocity (represented by data from the T2 IMU) in all movement planes, the BTW group participants increased their variability in the velocity of the trunk flexion and extension and in the velocity of the bicycle’s lateral oscillations (represented by data from the bicycle frame IMU). In the last observation phase, O3, the BTW children could not use their training wheels (like infants leaving a baby walker), so their center of gravity was no longer stable. Consequently, they were forced to explore how to regulate the postural control of their trunk, as well as controlling the bicycle.

As hypothesized, when comparing the movement variability of postural control between the two learning bikes, the BB provided greater postural variability in the child–bicycle system from the point of initial contact, in O1, as well as after practice, in O2. The BB group displayed greater variability in all movement planes for the head (represented by data from the vertex IMU) and for the trunk, as well as in all planes for the steering wheel. By not having any artificial support (i.e., the absence of the training wheels), it was more difficult to keep the BB balanced, even when feet were in contact with the ground, than to keep the BTW balanced. This is because, even when there are no feet in contact with the ground, the BTW bicycle does not fall and has small lateral oscillations. Besides this difference in task constraint, to ride the BB, the children had to self-propel with their feet on the ground. These relations of the children with the BB could result in the emergence of what has been termed a ‘neutral’ affordance landscape, supporting agency in exploring multiple BB cycle patterns, e.g., walking, running, hopping, and others [45,64]. Not long after the first contact with the BB, a child could simply push and maintain balance with the BB by gliding [45]. The BB supported the exploration of a greater variety of cycle patterns, leading the children to explore greater spatiotemporal variability in several segments and movement planes, which was reflected by the higher LyE values. The only exception was in the velocity of the bicycle frame’s lateral rotations, in which the BTW children displayed greater system variability compared to the BB group. This is a result that may not have been expected since the lateral wheels limited the amplitude of the bicycle’s lateral oscillations. However, even with lateral wheels, in tight curves or at higher speeds, the experimenters observed that the centrifugal force pushed the children’s trunk to move laterally, lifting one of the training wheels off the ground, resulting in a fall or an abrupt return of the wheel to the ground. This rapid oscillation produced by mechanical factors may have led to a higher LyE value, thus justifying this unexpected observation.

In comparing the measures of movement variability in the TB between both groups, only one statistically significant difference was found, with the BB group participants displaying higher variability in the velocity of the bicycle frame’s lateral rotations. Generally, these results show that the same foundational motor developmental capacities, in this case, the ability to cycle, can be achieved along different pathways, by using different training bicycles like the BB or BTW [9]. This observation is in line with what Waddington defined as the equifinality principle [65]. Interestingly, when analyzing movement variability in the two groups across all observations, in O1 and O2, the BB participants displayed significantly higher LyE values with lower standard deviations for all points and planes, except in the T2-O2-sagittal plane. But, this tendency was inverted in O3, when the BB children continued to display higher LyE mean values (with the only statistically significant difference being observed for bicycle frame’s rotation). Higher standard deviation values were observed for all points in time and movements, except in the vertex’s frontal plane. Despite having reduced the variability of their oscillation velocity when they transferred to the TB, between O2 and O3, the BB participants continued to display higher mean values of variability, especially in O3, when even higher standard deviation values were observed in this group. Considering that the standard deviation of LyE is a variability measure of variability itself, and that the BB participants showed greater success in learning to cycle than the BTW group, these findings imply that variability seems to have been used in a functional way as a performance solution and not as a problem.

These data corroborate the idea of Bernstein [66] that learning to coordinate complex actions, like riding a bicycle, is acquired by unfreezing, controlling, and mastering degrees of freedom (DOF), i.e., motor units, muscles, joints, limbs, movement axes, and planes. More recently, Berthouze and Lungarella [67] proposed an update to these ideas, verifying that the acquisition of coordination results from dynamic alternations between freezing and freeing DOFs, also arguing that the movement system needs to be perturbed to trigger these freezing and freeing mechanisms. Our hypothesis is that the functional properties of the BB may have the necessary structural level of perturbation to trigger the child–BB system for the emergence of diverse self-organized cycle patterns. However, this might require a complexification of the child’s postural variability, as was observed in the head and trunk IMUs’ values, in order to be attuned with the emergence of a greater BB functional variability, as expressed in bicycle frame and handlebar IMUs’ values.

Previous studies in movement science using LyE measures have reported that higher values have been associated with greater movement system variability and flexibility in responding more quickly to perturbations in order to better control balance [35,38]. Indeed, children from the BB group adapted more easily to the TB, being able to self-launch, ride for 10 m, and brake significantly more quickly than participants in the BTW group [13]. In contrast, the BTW participants needed more time to adapt and three of them were not able to cycle independently after practicing with that bike. As already noted by Burt et al. [10], when children training with a BB transit to a TB, they tend to reveal defensive responses with an increasing stiffness in the trunk and arms, which tends to impact their capacity to balance on the bicycle and, consequently, their ability to cycle independently. The lower-level variability afforded by training with the BTW did not propitiate the children with enough opportunities to achieve greater postural flexibility on the bike. In response, they ended up freezing their movement system’s DOF, which is typical in those in the early learning stage according to Bernstein [66].

The present findings are aligned with previous research on motor skill acquisition, particularly studies focusing on the development of balance and coordination in children. For instance, research on the acquisition of gross motor skills, such as running and jumping, has shown that variability in practice can enhance motor learning by promoting adaptability and variability in motor performance [68]. Similarly, studies on fine motor skills, such as handwriting and object manipulation, have demonstrated that diverse practice conditions can lead to more adaptive skill acquisition [69]. These parallels suggest that the principles of motor variability and adaptive learning observed in our study with balance bikes (BB) are consistent with broader motor learning theories.

This hypothesis was confirmed by the significantly lower levels of movement variability displayed by the non-riders in our study, especially in all movement planes analyzed at the BTW handlebar. The non-riders displayed lower mean values for movement variability measures than the independent riders in all analyzed segments (IMUs) and in all motion planes; see Table 2.

### 4.1. Pratical Aplications

The present results contribute to shedding more light on the reasons why the BB proves to be more efficient in learning to cycle compared to the more traditional BTW approach. The inherent exploration of balance and the provision of greater motor variability during the learning process are potential catalysts. These results, as well as those of previous studies which show that it is possible to learn to cycle independently through a BB from the age of three [9,13], have practical applications for both school and family contexts. Kindergarten teachers, primary school teachers, physical education teachers, coaches, parents, and family members who want to encourage children to learn to cycle independently at a young age should make a BB available to their children as early as possible [70], e.g., as soon as they have acquired independent walking skills. Providing this equipment in a school context contributes not only to learning to cycle but also to increasing motor skills, developing coordination skills such as balance and spatial orientation, and [20,40], not least, providing moments of sharing and fun between peers, contributing to the development of relational skills [71].

### 4.2. Strengths, Limitations, and Considerations for Future Studies

The results of this study highlight that higher levels of movement system variability may be one of the important reasons for greater efficiency in learning to cycle when using a BB. Indeed, this finding also provides insights about the most efficient motor learning strategy for learning to cycle, based on allowing learners to explore movement system variability. In studies of learning to cycle using different learning bikes, the LyE has presented itself as a useful nonlinear measure which is a reliable tool for studying human gait [35,36]. Here, we demonstrated its use for studying learning behaviors in cycling, reinforcing the utility and versatility of the LyE.

The small sample size of the present study may limit the generalizability of the findings. Despite the effort and methodological rigor in controlling several possibly confounding variables (e.g., body composition and motor competence), it was not possible to control others that may also have had some influence on learning, such as physical fitness levels or socio-economic factors. Future research needs to verify the current findings with a larger sample size, controlling more possible confounding variables and perhaps using a longer learning phase.

Balance ability in children can be influenced by intrinsic constraints such as age, motor competence, and prior experience. Research indicates that balance performance improves with age, as older children typically exhibit better postural control and stability due to more advanced neuromuscular development [72]. Higher motor competence is associated with better balance abilities, as children with greater motor skills can more effectively manage their body’s movements and maintain stability [73]. Prior experience with activities that challenge balance can also enhance a child’s ability to control their posture and respond to balance-related tasks [74]. Given these potential influences, it was crucial to evaluate these variables at the beginning of our intervention. Our assessments confirmed that there were no significant differences between the two groups in terms of age, motor competence, or prior cycling experience at the start of the study. This homogeneity reinforces the internal validity of our study, ensuring that the observed effects could be attributed to the intervention itself rather than pre-existing differences between the groups. Additionally, a child’s height can be considered an individual constraint and may affect their ability to learn how to ride a bike. Depending on the bike’s features, particularly the height of the seat from the ground, shorter children might not be able to sit with their feet comfortably touching the ground. This affects their ability to self-launch on the bike without assistance. This issue was observed in the current study, with the youngest children, aged three, representing a limitation of the study. It is recommended that future studies consider the suitability of bicycle dimensions for their sample and that bicycle manufacturers adjust their designs to better accommodate younger children.

Furthermore, there are several other nonlinear methodologies that could provide complementary insights into the coordination of movements, such as recurrence quantification analysis (RQA) [32], single scale entropy [34], and refined composite multiscale dispersion entropy (RCMDE) [33]. To gain a deeper understanding of the processes coordinating actions when learning to cycle, further investigations could be designed using and combining these nonlinear techniques. Incorporating these advanced methodologies can enhance our understanding of motor behavior and its underlying mechanisms, particularly in the context of motor development [75,76].

## 5. Conclusions

Our results revealed that using the BB allowed for the greater postural variability of the child–bicycle system formed during learning, compared to using the BTW. The BB technology allowed children to use it to explore skill adaptation, facilitating a faster to the TB. Movement system variability, viewed in more traditional theories as error or noise, needs to be re-evaluated as a part of a system’s adaptive dynamics, supporting the capacity to adapt to the environment and learn more efficiently. The variability captured by the LyE technique may reflect the freezing and freeing of DOFs, which afford the emergence of synergies between a child and bicycle, supporting the acquisition of the foundational motor skill of riding a bicycle.

The results of this study also provide empirical support, verifying the choice of the balance bike as an adequate instrument for learning to cycle autonomously. The data suggest that policy makers, cycling federations, coaches, educators, and parents should choose the BB over the BTW for children to learn to cycle.

## Figures and Tables

**Figure 1 jfmk-09-00266-f001:**
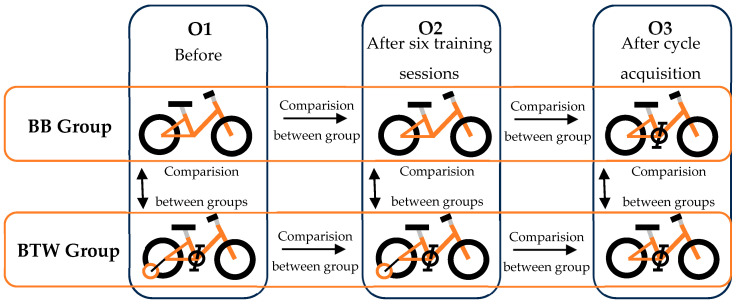
A presentation of the study design (2 groups × 3 moments), with the identification of the comparisons.

**Figure 2 jfmk-09-00266-f002:**
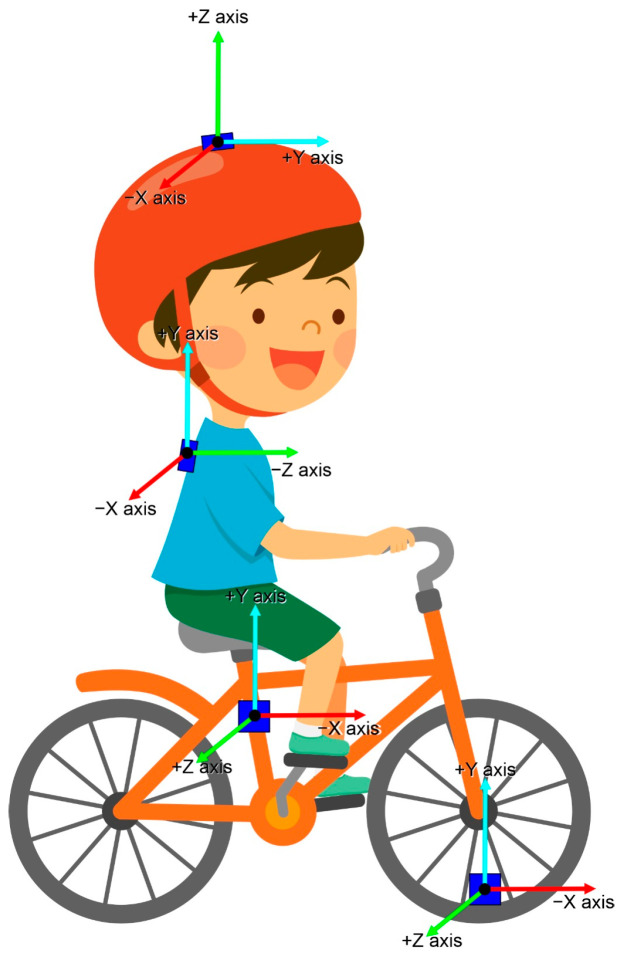
A graphical representation of the experimental setup with the sensor locations.

**Figure 3 jfmk-09-00266-f003:**
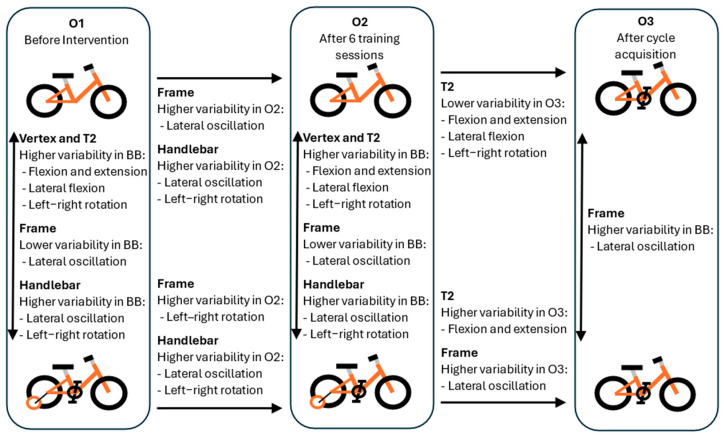
Abstract graph of comparisons of results between periods (O1, O2, O3) and between groups (BB, BTW).

**Table 1 jfmk-09-00266-t001:** Lyapunov descriptive statistics (M ± SD), in BB and BTW groups, for each IMU, movement plane, and evaluation moment (O1, O2, O3).

Group	IMU	Movement Plane	O1 M ± SD	O2 M ± SD	O3 M ± SD
BB	Vertex	Sagittal	58.82 ± 1.45	59.27 ± 1.03	58.50 ± 1.43
		Frontal	57.85 ± 1.14	57.72 ± 0.87	56.64 ± 1.26
		Transverse	56.51 ± 1.25	56.87 ± 1.04	55.63 ± 1.71
	T2	Sagittal	59.16 ± 1.51	58.48 ± 1.21	57.30 ± 0.99
		Frontal	56.61 ± 1.25	56.33 ± 0.77	55.39 ± 1.31
		Transverse	57.36 ± 0.80	57.16 ± 0.65	56.18 ± 1.63
	Bicycle frame	Frontal	54.32 ± 1.29	55.22 ± 0.72	55.86 ± 0.94
	Handlebar	Frontal	55.97 ± 1.08	57.33 ± 1.14	57.60 ± 1.48
		Transverse	55.84 ± 1.13	57.20 ± 1.14	57.72 ± 1.65
BTW	Vertex	Sagittal	56.60 ± 2.13	55.39 ± 1.35	58.41 ± 1.36
		Frontal	55.18 ± 1.76	57.51 ± 1.13	56.22 ± 1.98
		Transverse	53.91 ± 1.87	54.94 ± 1.35	54.37 ± 1.45
	T2	Sagittal	56.01 ± 1.82	56.28 ± 0.98	57.58 ± 0.78
		Frontal	52.58 ± 2.16	53.43 ± 1.21	54.55 ± 1.23
		Transverse	55.02 ± 1.87	55.97 ± 1.70	55.79 ± 0.98
	Bicycle frame	Frontal	57.52 ± 1.82	58.28 ± 1.49	55.01 ± 0.47
	Handlebar	Frontal	53.39 ± 2.12	55.35 ± 2.02	57.56 ± 1.10
		Transverse	53.54 ± 2.22	55.51 ± 1.66	57.68 ± 0.67

**Table 2 jfmk-09-00266-t002:** A table summarizing the significant differences between assessment moments within the same group.

Group	IMU	Movement Plane	O1 Versus O2 *p, r,* Direction Change	O2 Versus O3 *p, r,* Direction Change
BB	Vertex	Sagittal	ns	ns
		Frontal	ns	ns
		Transverse	ns	ns
	T2	Sagittal	ns	*p* = 0.034; *r* = 0.706; ↓
		Frontal	ns	*p* = 0.004; *r* = 0.775; ↓
		Transverse	ns	*p* = 0.043; *r* = 0.682; ↓
	Bicycle frame	Frontal	*p* = 0.035; *r* = 0.588; ↑	ns
	Handlebar	Frontal	*p* = 0.003; *r* = 0.748; ↑	ns
		Transverse	*p* = 0.04; *r* = 0.576; ↑	ns
BTW	Vertex	Sagittal	ns	ns
		Frontal	ns	ns
		Transverse	*p* = 0.025; *r* = 0.640; ↑	ns
	T2	Sagittal	ns	*p* = 0.020; *r* = 0.790; ↑
		Frontal	ns	ns
		Transverse	ns	ns
	Bicycle frame	Frontal	ns	*p* = 0.006; *r* = 0.865; ↑
	Handlebar	Frontal	*p* = 0.009; *r* = 0.713; ↑	ns
		Transverse	*p* = 0.012; *r* = 0.697; ↑	ns

Notes: ns—no statistical significance; ↑—significant increase between evaluations; ↓—significant decrease between evaluations.

**Table 3 jfmk-09-00266-t003:** A table summarizing the significant differences between groups within the same assessment moments.

Groups	IMU	Movement Plane	O1	O2	O3 (in TB)
BB versus BTW	Vertex	Sagittal	*p* = 0.008; *r* = 0.541; ↑	*p* = 0.001; *r* = 0.650; ↑	ns
		Frontal	*p* < 0.001; *r* = 0.689; ↑	*p* < 0.001; *r* = 0.792; ↑	ns
		Transverse	*p* = 0.001; *r* = 0.653; ↑	*p* = 0.008; *r* = 542; ↑	ns
	T2	Sagittal	*p* < 0.001; *r* = 0.771; ↑	*p* < 0.001; *r* = 0.720; ↑	ns
		Frontal	*p* < 0.001; *r* = 0.771; ↑	*p* < 0.001; *r* = 0.833; ↑	ns
		Transverse	*p* < 0.001; *r* = 0.654; ↑	*p* = 0.034; *r* = 0.443; ↑	ns
	Bicycle frame	Frontal	*p* < 0.001; *r* = 0.730; ↓	*p* < 0.001; *r* = 0.814; ↓	*p* = 0.048; *r* = 0.519; ↑
	Handlebar	Frontal	*p* = 0.001; *r* = 0.630; ↑	*p* = 0.008; *r* = 0.538; ↑	ns
		Transverse	*p* = 0.005; *r* = 0.570; ↑	*p* = 0.009; *r* = 0.530; ↑	ns

Notes: ns—no statistical significance; ↑—value significantly superior in BB group; ↓—value significantly inferior in BB group.

**Table 4 jfmk-09-00266-t004:** Lyapunov descriptive statistics of independent and non-independent riders, for each IMU, for movement planes and first (O1) and second (O2) periods.

BTW Group	IMU	Movement Plane	O1 M ± SD	O2 M ± SD
Independent Riders of BTW group	Vertex	Sagittal	56.72 ± 2.41	57.83 ± 1.18
		Frontal	55.08 ± 1.96	55.25 ± 1.30
		Transverse	54.07 ± 1.92	55.51 ± 1.43
	T2	Sagittal	56.14 ± 1.27	56.38 ± 1.07
		Frontal	52.48 ± 2.4	53.57 ± 1.37
		Transverse	55.04 ± 2.12	56.21 ± 1.65
	Bicycle frame	Frontal	57.43 ± 1.90	58.40 ± 1.66
	Handlebar	Frontal	53.96 ± 2.00	56.32 ± 1.12
		Transverse	54.18 ± 2.04	56.30 ± 0.98
Non-Independent Riders of BTW group	Vertex	Sagittal	56.28 ± 1.43	56.65 ± 0.26
		Frontal	55.43 ± 1.39	54.09 ± 1.31
		Transverse	53.49 ± 2.05	55.08 ± 1.29
	T2	Sagittal	55.65 ± 1.00	56.02 ± 0.80
		Frontal	52.84 ± 1.70	53.05 ± 0.69
		Transverse	54.98 ± 1.37	55.30 ± 1.99
	Bicycle frame	Frontal	57.78 ± 1.92	57.97 ± 1.03
	Handlebar	Frontal	51.86 ± 1.93	52.74 ± 1.45
		Transverse	51.83 ± 2.04	53.41 ± 1.16

## Data Availability

Data availability is possible upon request and with the approval of an institutional ethics committee.

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
