# Peer review of "Learning to Cycle: Why Is the Balance Bike More Efficient than the Bicycle with Training Wheels? The Lyapunov’s Answer"

_jfmk, 2024, doi:10.3390/jfmk9040266_

Round 1

Reviewer 1 Report

Comments and Suggestions for Authors

ABSTRACT

I sense some discrepancy here. On one hand, it is said that the balance bike is better than the TW, while on the other hand the authors said that they are going to explain why.  Has the efficacy of the balance not been already proved? This should be cleared out.

INTRODUCTION

Fundamental motor skills or milestones are well described in the literature. For instance, to pedal a three wheel bike is considered a motor development achievement by the CDCs, but not riding a bike. The authors cite reference 1 for supporting this statement. However, in this study, reading a bike is not mentioned. Hulteen et al mention that cycling (which is not exactly riding a bike according to my previous commentary on the CDCs milestones) is not a fundamental motor skill. Instead, they mention cycling, swimming  as other skills that contribute to physical activity. Therefore, a more solid background to justify the importance of learning how to ride a bike is needed in the introduction.

As a specialist in motor development who has taught to ride a bike to my own two sons, using both a BB and TW I can assure that TW are far better than BB, at least this was the case for my two children. What am I trying to say is that there seems to be only one investigation on this topic, so scientific evidence is very scarce. Therefore, the authors should justify their study on the bases of lack of research, instead of proclaiming that BB is better than TW, since most of the commentaries that they made in this regards seem to come more from speculation than from scientific background. BB might increase dynamic balance, but children do not learn how to break or how to keep balance while propelling themselves (they keep balance gaining advantage on the inertia that they create). On top of that, BB makes the children to break with their shoes, and I really can assure you that BB must be a creation of shoemakers and shoe stores, since I never seen so many shoes worn out in so little time because the use of BB in my life. On the other hand, TW are far better than BB because when the time is right, you can change the longitude of the TW without letting the children know. This allow for riding a bike with only one TW or sometimes non TW touching the ground and providing stabilization. The real deal in this process of learning to ride a bike is losing fear and gaining trust.

In closing, a better and not so biased justification towards BB might better suit the aim of this research.

METHDOS

Background information on the cycling camp is needed. What kind of children can take part in it? Is it a free camp?

It seems that children were not allocated randomly. Moreover, none information is provided regarding their basal motor skills, which are strongly linked to abilities like cycling. The authors merely state that children did not differ in motor competence and cite a previous investigation. Therefore, it is not known how motor skills were assessed, not the main values obtained in each group on motor competence.  Maturation level has been omitted too. Moreover, it is said that no differences were found on body composition, but this does not necessarily mean that there were not differences regarding height, or regarding weight. These limitations jeopardize the validity of the findings of this research.

Lines 130- 131, does this assumption mean that the children had never ride a bike until they entered the camp? Please, clarify this since previous experience on biking is crucial for the purpose of this research.

Line 136, it seems that height was a problem in three children in the BB group, but only in one children in the TW group. How can this be possible if the authors affirmed that there were no differences in body composition (see my previous comment in this regard). Moreover, three children did not perform O3 due to being small. In this regard, height should be an inclusion criteria since it is clear a cofounding factor.

Stats

I understand that this study has mainly a biomechanical focus. However, it seems interesting to add data regarding intra and intergroup changes. After all, this was a program aimed at learning how to ride a bike. How many children learnt to ride a bike in each group successfully? This information is slightly mentioned in the results section, but a more in depth approach both in the stats and consequently in the results section, should be welcomed.

RESULTS

As stated before, a table showing basal socio-demographic characteristics of the participants plus motor skill levels should be provided.

One or two figures could be of hand, since biomechanical data sometimes is easier to interpret in a graphical way.

DISCUSSION

I do not see the similarities between using baby walkers and TW at all.

The center of gravity is mentioned as a factor for learning a bike. Center of gravity depends on height, as shorter children show better balance (because the CoG is closer to the ground). I think that the authors should discuss the importance of height/balance/ and learning how to ride a bike.

Line 376, learning to riding a bike first depends on level of fear and security (line 395 mentions defensive responses, which I agree with). Both outcomes have not been assessed in this research. In addition, automation is the key factor for learning complex motor skills, like biking. The children first achieve balance, and then they can concentrate in other abilities on the bike. The authors should clear state that if from this research can be stated that balance is better achieved by using a BB.

Strengths and limitations are absent. Paragraphs from 407 to 415 can not be considered as strong/weak points related to the research.

CONCLUSION

First sentence can be erased.

Lines 429-434. It is not clear what are the authors trying to conclude here. Is this a weakness of the methodology used?

Author Response

Response Letter – Reviewer 1

Dear Reviewer,

First of all, we would like to thank you for taking the time to review our manuscript. Your comments have allowed us to reflect in greater depth, and we strongly believe that they have contributed to the improvement of the manuscript.

To make it easier to read our responses and our changes to the manuscript, we have responded using blue.

ABSTRACT

I sense some discrepancy here. On one hand, it is said that the balance bike is better than the TW, while on the other hand the authors said that they are going to explain why.  Has the efficacy of the balance not been already proved? This should be cleared out.

Response: Thank you for your comment, we do recognise that the sentences were confusing, so we tried to clarify them.

Text modified: However, the balance bike (BB) has consistently been regarded as more efficient, as children require less time on this bike to successfully transition to a traditional bike (TB). The reasons for this greater efficiency remain unclear, but it is hypothesized that it is due to the immediate balancing requirements for learners. This study aimed to investigate the reasons why BB is more efficient than BTW for learning to cycle on a TB.

INTRODUCTION

Fundamental motor skills or milestones are well described in the literature. For instance, to pedal a three wheel bike is considered a motor development achievement by the CDCs, but not riding a bike. The authors cite reference 1 for supporting this statement. However, in this study, reading a bike is not mentioned. Hulteen et al mention that cycling (which is not exactly riding a bike according to my previous commentary on the CDCs milestones) is not a fundamental motor skill. Instead, they mention cycling, swimming  as other skills that contribute to physical activity. Therefore, a more solid background to justify the importance of learning how to ride a bike is needed in the introduction.

Response: Dear Reviewer, we present cycling not as a fundamental motor skill, but rather as a foundational motor skill according to the new conceptual model proposed by Hulteen et al [1]. In this article, the authors suggest replacing the term 'fundamental movement skills' with 'foundational movement skills,' which underpins a significant conceptual adaptation to broaden the scope of skills considered important for promoting physical activity and other positive health trajectories across the lifespan. The authors also provided examples for this new terminology (which you can find in the key points on the first page of the article) which included specifically riding a bicycle, I quote: "Examples include resistance training movements, swimming strokes, and riding a bicycle." Indeed, several studies corroborate that learning to cycle supports adherence to and maintenance of healthy and positive trajectories, particularly due to the beneficial effects on physical health, such as improved body composition, enhanced cardiorespiratory fitness  [2], and mental health, including increased socialization opportunities and the development of social skills [3,4]. For all of these reasons, previous studies considered learning to ride a bicycle is an important milestone [5-7].

We recognize that this presentation is neither clear nor in-depth and that it may confuse readers who are unfamiliar with the Hulteen et al. model [1], so we appreciate your comments and suggestions for improvement. We have introduced this presentation and clarification in the introduction, adding more references that support this model [8] and the benefits of cycling.

References:

Hulteen, R.M.; Morgan, P.J.; Barnett, L.M.; Stodden, D.F.; Lubans, D.R. Development of Foundational Movement Skills: A Conceptual Model for Physical Activity Across the Lifespan. Sports Medicine 2018, 48, 1533-1540, doi:10.1007/s40279-018-0892-6.

Ramírez-Vélez, R.; García-Hermoso, A.; Agostinis-Sobrinho, C.; Mota, J.; Santos, R.; Correa-Bautista, J.E.; Amaya-Tambo, D.C.; Villa-González, E. Cycling to School and Body Composition, Physical Fitness, and Metabolic Syndrome in Children and Adolescents. The Journal of Pediatrics 2017, 188, 57-63, doi:https://doi.org/10.1016/j.jpeds.2017.05.065.

Karabaic, L. Putting the Fun Before the Wonk: Using Bike Fun to Diversify Bike Ridership. 2016.

O'Brien, E.; Pickering, T.; Asmar, R.; Myers, M.; Parati, G.; Staessen, J.; Mengden, T.; Imai, Y.; Waeber, B.; Palatini, P.; et al. Working Group on Blood Pressure Monitoring of the European Society of Hypertension International Protocol for validation of blood pressure measuring devices in adults. Blood Press Monit 2002, 7, 3-17, doi:https://doi.org/10.1097/00126097-200202000-00002.

Zeuwts, L.; Deconinck, F.; Vansteenkiste, P.; Cardon, G.; Lenoir, M. Understanding the development of bicycling skills in children: A systematic review. Safety Science 2020, 123, 104562, doi:https://doi.org/10.1016/j.ssci.2019.104562.

Zeuwts, L.; Ducheyne, F.; Vansteenkiste, P.; D'Hondt, E.; Cardon, G.; Lenoir, M. Associations between cycling skill, general motor competence and body mass index in 9-year-old children. Ergonomics 2015, 58, 160-171, doi:10.1080/00140139.2014.961971.

Mercê, C.; Pereira, J.V.; Branco, M.; Catela, D.; Cordovil, R. Training programmes to learn how to ride a bicycle independently for children and youths: a systematic review. Physical Education and Sport Pedagogy 2021, 1-16, doi:10.1080/17408989.2021.2005014.

Kavanagh, J.A.; Issartel, J.; Moran, K. Quantifying cycling as a foundational movement skill in early childhood. Journal of Science and Medicine in Sport 2020, 23, 171-175, doi:https://doi.org/10.1016/j.jsams.2019.08.020.

As a specialist in motor development who has taught to ride a bike to my own two sons, using both a BB and TW I can assure that TW are far better than BB, at least this was the case for my two children. What am I trying to say is that there seems to be only one investigation on this topic, so scientific evidence is very scarce. Therefore, the authors should justify their study on the bases of lack of research, instead of proclaiming that BB is better than TW, since most of the commentaries that they made in this regards seem to come more from speculation than from scientific background. BB might increase dynamic balance, but children do not learn how to break or how to keep balance while propelling themselves (they keep balance gaining advantage on the inertia that they create). On top of that, BB makes the children to break with their shoes, and I really can assure you that BB must be a creation of shoemakers and shoe stores, since I never seen so many shoes worn out in so little time because the use of BB in my life. On the other hand, TW are far better than BB because when the time is right, you can change the longitude of the TW without letting the children know. This allow for riding a bike with only one TW or sometimes non TW touching the ground and providing stabilization. The real deal in this process of learning to ride a bike is losing fear and gaining trust.

In closing, a better and not so biased justification towards BB might better suit the aim of this research.

Response: I'm not questioning your personal experience with your two children, as you well known, each child has their own specificities and the tool/task that best suits the majority doesn't have to be the best alternative for all children. However, this was a personal experience and not a scientific study. Finally, I would like to emphasise that this study did not aim to assess fear or motivation. There is scientific literature pointing out that BTW is counterproductive because it doesn't allow the exploration of balance during learning, which triggers defensive reactions of freezing degrees of freedom when transitioning to the conventional bicycle [9-15]. There are several studies that explore the use of different learning bikes, pointing to the greater efficiency of the BB, including a systematic review [7], a retrospective study that emphasises that children who learn to cycle on the BB learn on average 1.81 years earlier than those using BTW [16], and also a semi-experimental study that applied an intervention programme with two groups, one using BB and the other using BTW, finding that the BB children needed significantly fewer sessions to learn all the milestones of cycling and the ability to cycle autonomously [17]. We have strengthened this justification and added more references from the literature to reinforce this issue.

References:

Cain, S.M.; Ulrich, D.; Perkins, N. Using Measured Bicycle Kinematics to Quantify Increased Skill as a Rider Learns to Ride a Bicycle. In Proceedings of the ASME 2012 5th Annual Dynamic Systems and Control Conference joint with the JSME 2012 11th Motion and Vibration Conference, Florida, USA, 2012.

Hauck, J.; Jeong, I.; Esposito, P.; MacDonald, M.; Hornyak, J.; Argento, A.; Ulrich, D.A. Benefits of Learning to Ride a Two-Wheeled Bicycle for Adolescents with Down Syndrome and Autism Spectrum Disorder. PALAESTRA 2017, 31.

Hawks, Z.; Constantino, J.N.; Weichselbaum, C.; Marrus, N. Accelerating Motor Skill Acquisition for Bicycle Riding in Children with ASD: A Pilot Study. J Autism Dev Disord 2020, 50, 342-348, doi:10.1007/s10803-019-04224-5.

MacDonald, M.; Esposito, P.; Hauck, J.; Jeong, I.; Hornyak, J.; Argento, A.; Ulrich, D.A. Bicycle Training for Youth With Down Syndrome and Autism Spectrum Disorders. Focus on Autism and Other Developmental Disabilities 2012, 27, 12-21, doi:10.1177/1088357611428333.

Temple, V.A.; Purves, P.L.; Misovic, R.; Lewis, C.J.; DeBoer, C. Barriers and Facilitators for Generalizing Cycling Skills Learned at Camp to Home. Adapted Physical Activity Quarterly 2016, 33, 48-65, doi:10.1123/apaq.2015-0040.

Ulrich, D.A.; Burghardt, A.R.; Lloyd, M.; Tiernan, C.; Hornyak, J.E. Physical activity benefits of learning to ride a two-wheel bicycle for children with Down syndrome: a randomized trial. Phys Ther 2011, 91, 1463-1477, doi:10.2522/ptj.20110061.

Burt, T.L.; Porretta, D.P.; Klein, R.E. Use of Adapted Bicycles on the Learning of Conventional Cycling by Children with Mental Retardation. Education and Training in Developmental Disabilities 2007, 42, 364–379.

Mercê, C.; Branco, M.; Catela, D.; Lopes, F.; Cordovil, R. Learning to Cycle: From Training Wheels to Balance Bike. International Journal of Environmental Research and Public Health 2022, 19, 1814, doi:10.3390/ijerph19031814.

Mercê, C.; Davids, K.; Catela, D.; Branco, M.; Correia, V.; Cordovil, R. Learning to cycle: a constraint-led intervention programme using different cycling task constraints. Physical Education and Sport Pedagogy 2023, 1-14, doi:10.1080/17408989.2023.2185599.

METHDOS

Background information on the cycling camp is needed. What kind of children can take part in it? Is it a free camp?

Response: Yes, participation in the study was completely voluntary and free of charge, and participants could leave the study if they wished without having to give reasons. In order to clarify this issue, we have added more information in the ‘Participants’ section.

It seems that children were not allocated randomly. Moreover, none information is provided regarding their basal motor skills, which are strongly linked to abilities like cycling. The authors merely state that children did not differ in motor competence and cite a previous investigation. Therefore, it is not known how motor skills were assessed, not the main values obtained in each group on motor competence.  Maturation level has been omitted too. Moreover, it is said that no differences were found on body composition, but this does not necessarily mean that there were not differences regarding height, or regarding weight. These limitations jeopardize the validity of the findings of this research.

Response: It is worth noting that the aim of this study was to investigate whether functional variability could be one of the reasons for BB's greater efficiency. The data used to fulfill this aim was collected simultaneously with the application of the Learning to Cycle program. This program has already had its results by comparing the number of sessions required to achieve independent cycling, as well as the analysis of the influence of the participants' constraints (e.g., motor competence, body composition, previous experience) discussed and presented in a previous study. Considering this aspect and that these variables were not the objective under study for the present manuscript, we did not explore them in depth in the first version of the manuscript. We recognize that this information may not be clear, so we have added a more detailed explanation in section ‘2.1. Study Design’. This way, if readers want to know more about variables that were not the focus of this study, they can consult the other reference.

Nevertheless, we recognized that the “participants” section of the manuscript was not very in-depth. We have clarified the stratified randomization process used to form the two groups. The exclusion criteria (highlighted in orange) involved excluding all children who, prior to the intervention, already knew how to cycle independently. We specified all the variables analyzed to demonstrate the absence of significant differences between the two groups. Additionally, in the "Data Collection and Protocols" section, we detailed all the variables collected and the protocols followed to ensure the initial conditions of the two groups were similar at the beginning of the intervention (i.e., weight, height, BMI, BMI’s percentile, motor competence and previous cycling experience). Considering that all the children were between 3 and 7 years old, they were all in the prepubertal stage, so we did not consider maturation as a possible confounding variable.

Lines 130- 131, does this assumption mean that the children had never ride a bike until they entered the camp? Please, clarify this since previous experience on biking is crucial for the purpose of this research.

Response: We apologize, but the lines you indicated for this comment corresponded to an image, so we could not clearly identify it. As stated in the previous comment, prior experience with the bicycle was assessed through a questionnaire administered to the guardians. Around 90% of the sample reported having no prior experience with bicycles, while some indicated having used a balance bike 1 to 6 times per year. When we compared the presence of this experience between groups, the difference was not significant, ensuring similar initial conditions between groups. We have added information about the control of this variable in the "Participants" and "Data Collection and Protocols" sections (changes highlighted in green).

Line 136, it seems that height was a problem in three children in the BB group, but only in one children in the TW group. How can this be possible if the authors affirmed that there were no differences in body composition (see my previous comment in this regard). Moreover, three children did not perform O3 due to being small. In this regard, height should be an inclusion criteria since it is clear a cofounding factor.

Response: Thank you for your comment. Indeed, height can be an individual’s constraint that influences their ability to learn to cycle. We reconfirmed that there were no statistically significant differences in body composition between groups, including height. However, this does not rule out the presence of two children in the BB group whose height prevented them from reaching the ground. We understand and acknowledge in the manuscript that this is a limitation, but we do not agree that the correct approach would have been to exclude the shorter children. If we had chosen that option, we would have excluded the younger children (aged 3), thereby limiting the study's potential. With the option we chose, we were able to support the possibility of successfully learning to cycle at a young age. Recognizing the value and relevance of your comment, we have explicitly mentioned this situation in section "4.2. Strengths, limitations, and considerations for future studies," identifying it as a limitation and suggesting that future studies consider the suitability of bicycle dimensions for the sample.

Stats

I understand that this study has mainly a biomechanical focus. However, it seems interesting to add data regarding intra and intergroup changes. After all, this was a program aimed at learning how to ride a bike. How many children learnt to ride a bike in each group successfully? This information is slightly mentioned in the results section, but a more in depth approach both in the stats and consequently in the results section, should be welcomed.

Response: Thank you for the comment. We have provided more details on this issue in section "3.3. Comparisons with children who did not acquire the skill of cycling independently."

RESULTS

As stated before, a table showing basal socio-demographic characteristics of the participants plus motor skill levels should be provided.

Response: We appreciate the comment. As previously stated, the collection of these biomechanical data took place during the implementation of the Learning to Cycle program. The characterization data and variables representing individual constraints have already been discussed and presented in another published study [17]. We have clarified this information in the manuscript, clearly identifying the reference so that readers can, if they wish, read more about these variables that were not the direct focus of the present manuscript.

Reference:

Mercê, C.; Davids, K.; Catela, D.; Branco, M.; Correia, V.; Cordovil, R. Learning to cycle: a constraint-led intervention programme using different cycling task constraints. Physical Education and Sport Pedagogy 2023, 1-14, doi:10.1080/17408989.2023.2185599.

One or two figures could be of hand, since biomechanical data sometimes is easier to interpret in a graphical way.

Response: Thank you for the suggestion. Indeed, some biomechanical data are more easily interpreted in graphical form. However, considering the number of variables analyzed, including data collected at three different times, with two groups, and for the three axes of movement, we felt that the graphs would become too extensive and might overwhelm the reader rather than aid in interpretation. Instead, we opted to include two schematic images: one that simplifies the study design (Figure 1) and another that highlights the significant differences and their direction in both intra- and inter-group comparisons (Figure 2).

DISCUSSION

I do not see the similarities between using baby walkers and TW at all.

Response: We respect your opinion. In light of the conceptual and theoretical model of the study, non-linear pedagogy, which advocates for introducing variability during the learning process so that the learner can explore various motor solutions, self-organize, and acquire the new motor skill with adaptability. We compared the use of baby walkers to BTW because both artificially limit the exploration of this variability, i.e., the exploration of balance, by imposing artificial physical constraints, whether it be the seat itself or the extra training wheels. By using both instruments, children face physical limitations that prevent them from exploring their balance.

The center of gravity is mentioned as a factor for learning a bike. Center of gravity depends on height, as shorter children show better balance (because the CoG is closer to the ground). I think that the authors should discuss the importance of height/balance/ and learning how to ride a bike.

Response: Biomechanically, the ability to balance is affected by several factors [18]: the vertical distance from the center of gravity to the ground, where theoretically shorter children would have an advantage; mass, where theoretically heavier children would have an advantage (which is not observed in numerous studies with overweight and obese children); base of support; and friction, which remained identical in both groups. Maintaining balance is influenced by the interaction of various variables, so stating that shorter children will have better balance can be extremely reductive. Shorter children may imply younger, immature children with lower motor competence, which in turn brings additional difficulty in balancing. We appreciate your comment, which led us to further reflection; however, this was not the objective of the study, so we may consider it in future studies.

Reference:

Hall, S. Basic Biomechanics, 9th ed.; McGraw Hill: New York, 2022.

Line 376, learning to riding a bike first depends on level of fear and security (line 395 mentions defensive responses, which I agree with). Both outcomes have not been assessed in this research. In addition, automation is the key factor for learning complex motor skills, like biking. The children first achieve balance, and then they can concentrate in other abilities on the bike. The authors should clear state that if from this research can be stated that balance is better achieved by using a BB.

Response: We acknowledge that fear can limit a child's ability to learn cycling, while motivation can facilitate it. However, in this study, these variables were not directly measured or analyzed. We can share some anecdotal observations: we empirically noted that children in the BTW group, when transitioning to a conventional bike, exhibited significant stiffness throughout their upper limbs (i.e., shoulder, elbow, and wrist). Some of them even verbally expressed that "the bike without training wheels feels like it's going to fall." Although we did not assess fear directly, we evaluated motor variability. The freezing of degrees of freedom and movements (possibly due to fear) leads to lower variability values. This study demonstrated that the BB promotes greater motor variability during the learning process compared to the BTW, which may have contributed to a more efficient transition to the conventional bike. Following your comment, we sought to clarify this aspect.

Strengths and limitations are absent. Paragraphs from 407 to 415 can not be considered as strong/weak points related to the research.

Response: Thank you for your comments. We have thoroughly reformulated the section ‘4.2 Strengths, limitations and considerations for future studies’, the changes are in orange and green as they also correspond to comments from other reviewers.

CONCLUSION

First sentence can be erased.

Response: Alteration done.

Lines 429-434. It is not clear what are the authors trying to conclude here. Is this a weakness of the methodology used?

Response: Not at all. The identified section presents suggestions for future studies. We reinforced the introduction by presenting and justifying the choice of the LyE for the specific objective (information highlighted in green following another reviewer's suggestion), acknowledging that there are other non-linear methodologies that, while not as well-suited, could provide additional complementary information to this analysis. Non-linear methods encompass multiple methodologies that are suited to different objectives and types of data signals. In some cases, it may be beneficial to combine various methods to achieve a more comprehensive and in-depth understanding of the objective. In this regard, and recognizing that the message may not have been clear in the section identified by the esteemed reviewer, we have proceeded to improve and clarify it.

Reviewer 2 Report

Comments and Suggestions for Authors

Dear Editor,

Thank you for inviting me to review manuscript ID jfmk-3337315 entitled "Learning to Cycle: why is the balance bike more efficient than the bicycle with training wheels? The Lyapunov's answer." This study investigates the mechanisms behind the superior effectiveness of balance bikes compared to training wheels for learning to cycle, using nonlinear analysis of movement variability. The main finding suggests that balance bikes allow greater functional variability in the child-bicycle system during learning, facilitating more efficient skill acquisition. Overall, this is an interesting and well-conducted study that makes a valuable contribution to understanding motor learning in cycling skills. I recommend minor revisions before publication.

General Comments

1.   Theoretical Framework

·     The connection between movement variability and motor learning could be strengthened with additional theoretical background

·     The rationale for using Lyapunov exponents should be more explicitly justified

·     Consider discussing alternative nonlinear measures that could complement the analysis

2.   Methodology

·     The sampling procedures and participant selection criteria need more detail

·     The reliability and validity of the IMU measurements should be addressed

·     Statistical analysis methods could benefit from more thorough explanation

3.   Results & Discussion

·     Some findings need clearer interpretation regarding practical implications

·     The discussion could better integrate findings with existing motor learning literature

·     Consider addressing limitations more comprehensively

Specific Comments

Introduction

1.   Expand on how movement variability relates to skill acquisition

2.   Clarify the hypothesized mechanisms behind balance bike effectiveness

3.   Include more recent references on nonlinear analysis in motor development

Methods

4.   Provide details on IMU calibration and data quality control

5.   Explain the rationale for choosing specific measurement locations

6.   Describe participant familiarization procedures more thoroughly

Results

7.   Include effect sizes for all statistical comparisons

8.   Present individual variation data alongside group means

9.   Consider adding qualitative observations of movement patterns

Discussion

10.          Elaborate on the practical implications for physical education

11.          Compare findings with similar studies in other motor skills

12.          Discuss how age might influence the observed effects

Tables/Figures

13.          Add error bars to figures

14.          Improve table formatting for clarity

15.          Consider adding a schematic of the experimental setup

This manuscript addresses an important topic in motor development and provides novel insights through sophisticated analysis. With the suggested revisions, it will make a valuable contribution to the field.

Thank you for the opportunity to review this interesting work.

Best regards,

The reviewer

Author Response

Response Letter – Reviewer 2

Dear Reviewer,

First of all, we would like to thank you for taking the time to review our manuscript. Your comments have allowed us to reflect in greater depth, and we strongly believe that they have contributed to the improvement of the manuscript.

To make it easier to read our responses and our changes to the manuscript, we have responded using green.

Dear Editor,

Thank you for inviting me to review manuscript ID jfmk-3337315 entitled "Learning to Cycle: why is the balance bike more efficient than the bicycle with training wheels? The Lyapunov's answer." This study investigates the mechanisms behind the superior effectiveness of balance bikes compared to training wheels for learning to cycle, using nonlinear analysis of movement variability. The main finding suggests that balance bikes allow greater functional variability in the child-bicycle system during learning, facilitating more efficient skill acquisition. Overall, this is an interesting and well-conducted study that makes a valuable contribution to understanding motor learning in cycling skills. I recommend minor revisions before publication.

Response: Thank you very much for your appreciation, it's a pleasure for the whole team to realise that you liked our manuscript.

General Comments

  1. Theoretical Framework
  • The connection between movement variability and motor learning could be strengthened with additional theoretical background

Response: Thank you for your comment. As suggested, we have added more theoretical background, specifically highlighting variability as a crucial element of nonlinear pedagogy. This addition reinforces and clarifies its importance in motor skill acquisition and learning.

  • The rationale for using Lyapunov exponents should be more explicitly justified

Response: There are several non-linear analysis methods that could be applied to treat these data, namely recurrence quantification analysis (RQA) [1], which evaluates the recurrence of dynamic states in time series; or the single scale entropy, which can be used as a measure of uncertainty and irregularity of time series [2,3]. However, considering the purpose of the current study, which is to specifically analyze motor variability; and its specifications, including periodic data form angular velocities, the most suitable nonlinear technique consists in the largest Lyapunov exponent (LyE) [4,5]. The LyE is probably the most popular nonlinear methods used to assess stability and variability [4,6]. This method is widely used in the analysis of biological systems because they offer a deeper understanding of neuromotor control of movement. It is very sensitive to initial conditions and the divergence of trajectories in dynamic systems, providing a robust measure of system stability and variability.

This was the rationale that led us to choose and apply LyE, justifying it with appropriate literature that we identified in the manuscript. This entire explanation was added to the manuscript. We consider that it went deeper and added robustness to the choice of method used, so we would like to thank you for your suggestion.

References:

Recurrence Quantification Analysis: Theory and Best Practices; Webber, C.L., Marwan, N., Eds.; Springer: Switzerland, 2015.

Azami, H.; Rostaghi, M.; Abásolo, D.; Escudero, J. Refined Composite Multiscale Dispersion Entropy and its Application to Biomedical Signals. IEEE Transactions on Biomedical Engineering 2017, 64, 2872-2879, doi:10.1109/TBME.2017.2679136.

3Yentes, J.M.; Raffalt, P.C. Entropy Analysis in Gait Research: Methodological Considerations and Recommendations. Annals of Biomedical Engineering 2021, 49, 979-990, doi:10.1007/s10439-020-02616-8.

4Kędziorek, J.; Błażkiewicz, M. Nonlinear Measures to Evaluate Upright Postural Stability: A Systematic Review. Entropy 2020, 22, 1357.

Mehdizadeh, S. The largest Lyapunov exponent of gait in young and elderly individuals: A systematic review. Gait & Posture 2018, 60, 241-250, doi:https://doi.org/10.1016/j.gaitpost.2017.12.016.

da Costa, C.S.; Batistão, M.V.; Rocha, N.A. Quality and structure of variability in children during motor development: a systematic review. Res Dev Disabil 2013, 34, 2810-2830, doi:10.1016/j.ridd.2013.05.031.

  • Consider discussing alternative nonlinear measures that could complement the analysis

Response: Thank you very much for your suggestion. Following the previous comment, we ended up adding a small reflection on other non-linear methods in the introduction. Considering the specificity of the study objective, we strongly believe that the most appropriate method is LyE. Following your comment, we emphasized in the recommendations the potential for future studies to use other non-linear techniques (providing suggestions) and even to combine several methods for a more in-depth analysis.

  1. Methodology
  • The sampling procedures and participant selection criteria need more detail

Response: Thank you for your suggestion. Indeed, this section of the manuscript was not very in-depth. We have clarified the stratified randomization process used to form the two groups. The exclusion criteria (highlighted in orange) involved excluding all children who, prior to the intervention, already knew how to cycle independently. We specified all the variables analyzed to demonstrate the absence of significant differences between the two groups. Additionally, in the "Data Collection and Protocols" section, we detailed all the variables collected and the protocols followed to ensure the initial conditions of the two groups were similar at the beginning of the intervention (i.e., weight, height, BMI, BMI’s percentile, motor competence and previous cycling experience).

  • The reliability and validity of the IMU measurements should be addressed

Response: Thank you for your comment. We have explained and strengthened these issues in the ‘Data Collection and Protocols’ section in orange.

  • Statistical analysis methods could benefit from more thorough explanation

Response: Thanks for the suggestion, we improved the section and added the display of the effect size calculation, reinforcing it with a literature reference.

  1. Results & Discussion
  • Some findings need clearer interpretation regarding practical implications

Response: Thank you very much for your comment, we've added the ‘Practical Implications’ section to the discussion.

  • The discussion could better integrate findings with existing motor learning literature

Response: Thank you very much for your suggestion. We have tried to integrate our results with the motor development literature by adding a new paragraph to the discussion.

  • Consider addressing limitations more comprehensively

Response: As suggested, we seek to add more information in the "Strengths, limitations and considerations for future studies" section, presenting the limitations in a more comprehensive way. Namely, through recognition of the sample size, which should be carefully considered when generalizing the data; as well as the recognition that, although we sought to control the main confounding variables, it is necessary to continue identifying and continuously controlling them in future studies.

Specific Comments

Introduction

  1. Expand on how movement variability relates to skill acquisition

Response: In order to fulfill this suggestion, we added more theoretical background, which explains the role of variability in motor acquisition and learning (also responding to the 1st comment regarding Theoretical Framework presented above).

  1. Clarify the hypothesized mechanisms behind balance bike effectiveness

Response: As requested, we clarify and justify the hypothesized mechanisms behind the balance bike's effectiveness. We chose to place this presentation in the last paragraph of the discussion to better linking the objectives to the hypotheses raised.

  1. Include more recent references on nonlinear analysis in motor development

Response: As suggested, we added more recent references to nonlinear analysis in motor development. We made this addition both in the introduction and in future recommendations, adding the following references:

Recurrence Quantification Analysis: Theory and Best Practices; Webber, C.L., Marwan, N., Eds.; Springer: Switzerland, 2015.

Azami, H.; Rostaghi, M.; Abásolo, D.; Escudero, J. Refined Composite Multiscale Dispersion Entropy and its Application to Biomedical Signals. IEEE Transactions on Biomedical Engineering 2017, 64, 2872-2879, doi:10.1109/TBME.2017.2679136.

Yentes, J.M.; Raffalt, P.C. Entropy Analysis in Gait Research: Methodological Considerations and Recommendations. Annals of Biomedical Engineering 2021, 49, 979-990, doi:10.1007/s10439-020-02616-8.

Moreno, F.J.; Caballero, C.; Barbado, D. Editorial: The role of movement variability in motor control and learning, analysis methods and practical applications. Frontiers in Psychology 2023, 14, doi:10.3389/fpsyg.2023.1260878.

Getchell, N.; Schott, N.; Brian, A. Motor Development Research: Designs, Analyses, and Future Directions. Journal of Motor Learning and Development 2020, 8, 410-437, doi:10.1123/jmld.2018-0029.

Methods

  1. Provide details on IMU calibration and data quality control

Response: Thank you for your comment. We have explained the sensor calibration procedures, reinforcing them with a literature reference in the ‘Data Collection and Protocols’ section in orange.

  1. Explain the rationale for choosing specific measurement locations

Response: We briefly presented this rationale earlier to avoid making the methodology section too dense. Following your request, we have detailed the rationale in the 'Data Collection and Protocols' section. The chosen sites provide a comprehensive analysis of both the rider's body movements and the bicycle's dynamics, allowing for a detailed understanding of the child-bicycle system during the learning process. Additionally, we have provided specific justifications for the placement of each IMU.

  1. Describe participant familiarization procedures more thoroughly

Response: To ensure that participants were comfortable and familiar with the study procedures and equipment, a thorough familiarization process was conducted prior to the data collection. This process included the following steps: i) introduction to the task, presented as a game; ii) equipment familiarization, participants were introduced to the customized vest with the inertial measurement units (IMUs), this equipment was presented as the superhero outfit that was going to collect some information about the game they were playing; iii) equipment exploration, prior to mount the bicycle the children were invited to walk and run a lite with the customized vest in order to get used to the feel of the equipment; iv) safety and comfort checks, throughout the familiarization process, investigators conducted regular checks to ensure that the equipment was properly fitted and that participants were comfortable.

By implementing these familiarization procedures, we aimed to ensure that participants were well-prepared and comfortable with the study procedures, thereby enhancing the quality and reliability of the data collected. As suggested, we added this information in the “2.4. Data Collection and Protocols” section.

Results

  1. Include effect sizes for all statistical comparisons

Response: Dear reviewer, we calculated the effect size r [7] for all comparisons with significant differences. These results are presented in text and in tables 2 and 3.

Reference:

Field, A. Discovering Statistics Using IBM SPSS Statistics; SAGE Publications: 2013.

  1. Present individual variation data alongside group means

Response: Dear reviewer, the presentation of central statistical values (i.e., the mean) is always accompanied by the presentation of the dispersion value (i.e., standard deviation), as can be seen in Tables 1 and 4."

  1. Consider adding qualitative observations of movement patterns

Response: Thank you very much for your suggestion, in fact this is an aspect that we have already considered carrying out in future studies. However, unfortunately, as this was not something we included in the data collections, we will not be able to add this information now.

Discussion

  1. Elaborate on the practical implications for physical education

Response: Thank you for the suggestion. Indeed, these results, along with previous literature, have significant practical applications in both school and family contexts. The early provision of balance bikes (BB) in these settings can accelerate the learning of independent cycling, while simultaneously enhancing motor literacy, coordination skills, and relational skills. Following your suggestion, and to improve the discussion, we have added a 'Practical Applications' section. In this section, we have detailed and explained all these applications, reinforcing them with various references from current literature.

  1. Compare findings with similar studies in other motor skills

Response: As suggested, we have tried to integrate our results with more motor development literature, particularly in other motor skills such as running, jumping or handwriting, by adding a new paragraph to the discussion.

  1. Discuss how age might influence the observed effects

Response: Thank you very much for your suggestion. In fact, there are several factors that can influence a child's ability to balance and stabilise, including their age. The influence of these factors such as age, motor skills or previous experience was not the aim of the study. Nevertheless, we recognised their importance and assessed them to ensure similarity between the groups at the start of the intervention. Only with this methodological care, which reinforces the internal validity of the study, could we accurately compare the evolution of variability during the process of learning to cycle in two groups with different learning cycles. We have added this reflection, supported by literature references, to the discussion, specifically in section ‘4.2. Strengths, limitations and considerations for future studies’.

Tables/Figures

  1. Add error bars to figures

Response: Dear reviewer, our figures consist of diagrams designed to help you understand: the study design, namely the organization of the groups and assessment times (Figure 1), the presence and direction of significant differences in each sensor by group and between groups (Figure 2). Due to this specificity, we are unable to add error bars.

  1. Improve table formatting for clarity

Response: To simplify the interpretation of the tables, we included only statistically significant values, using symbols to streamline data presentation. We also adhered to the journal's specific guidelines for table formatting. Following your comment, we adjusted the table sizes, particularly Table 4, to make them more visually appealing. If you have any further suggestions for improvement, please let us know, and we will be happy to make the necessary changes.

  1. Consider adding a schematic of the experimental setup

Response: Thank you for the suggestion. In accordance with it, we have added Figure 3, which visually represents the experimental setup.

This manuscript addresses an important topic in motor development and provides novel insights through sophisticated analysis. With the suggested revisions, it will make a valuable contribution to the field.

Thank you for the opportunity to review this interesting work.

Best regards,

Response: Once again, a heartfelt thank you for all your positive comments and suggestions for improvement.

Reviewer 3 Report

Comments and Suggestions for Authors

The introduction provided sufficient background and demonstrated a need for this research. The manuscript was well-organized. Some additional information is needed in the methods section. Some of the narrative in the results section could be written more concisely. The discussion was coherent with a discussion that explained the results and examined previous/related research. Parts of the conclusion need minor revision to make it more conservative. There are some word choice and tense errors, with a few grammatical errors. See specific comments in the PDF.

Author Response

Response Letter – Reviewer 3

Dear Reviewer,

First of all, we would like to thank you for taking the time to review our manuscript. Your comments have allowed us to improve and clarify the manuscript.

To make it easier to read our responses and our changes to the manuscript, we have responded using orange.

The introduction provided sufficient background and demonstrated a need for this research. The manuscript was well-organized. Some additional information is needed in the methods section. Some of the narrative in the results section could be written more concisely. The discussion was coherent with a discussion that explained the results and examined previous/related research. Parts of the conclusion need minor revision to make it more conservative. There are some word choice and tense errors, with a few grammatical errors. See specific comments in the PDF.

Response: We sincerely appreciate and thank you for all your comments. By addressing the specific points below, we believe we have covered all the general feedback provided.

Reviewer: highlighted words have wrong tense, they should be "would afford" and "would be"; re-reviewing the manuscript for other word tense errors should be done

Response: Thank you for noticing, we have changed it accordingly.

Reviewer: what were the explicit exclusion criteria?

Response: The exclusion criterion consisted of the ability to cycle independently prior to the intervention. We realised that this criterion might not be entirely clear, so we added another sentence to explain it clearly.

Please provide references on the tested reliability and validity of the IMUs; reliability verification is particularly important because of the test-retest nature of this project;

Response: Thank you for your valuable comments and suggestions. To address your concern regarding the reliability and validity of the inertial measurement units (IMUs), we have included several references that demonstrate the robustness of these devices in various movement contexts, including cycling. We also add more text and present in a deeper way the advantages of using IMUs for collecting biomechanic data.

Additional References:

Camomilla, V.; Bergamini, E.; Fantozzi, S.; Vannozzi, G. Trends Supporting the In-Field Use of Wearable Inertial Sensors for Sport Performance Evaluation: A Systematic Review. Sensors 2018, 18, 873, doi:10.3390/s18030873.

Winter, L.; Bellenger, C.; Grimshaw, P.; Crowther, R.G. Analysis of Movement Variability in Cycling: An Exploratory Study. Sensors 2023, 23, 4972.

Zeng, Z.; Liu, Y.; Hu, X.; Tang, M.; Wang, L. Validity and Reliability of Inertial Measurement Units on Lower Extremity Kinematics During Running: A Systematic Review and Meta-Analysis. Sports Medicine - Open 2022, 8, 86, doi:10.1186/s40798-022-00477-0.

Kobsar, D.; Charlton, J.M.; Tse, C.T.F.; Esculier, J.-F.; Graffos, A.; Krowchuk, N.M.; Thatcher, D.; Hunt, M.A. Validity and reliability of wearable inertial sensors in healthy adult walking: a systematic review and meta-analysis. Journal of NeuroEngineering and Rehabilitation 2020, 17, 62, doi:10.1186/s12984-020-00685-3.

Clemente, F.; Badicu, G.; Hasan, U.; Akyildiz, Z.; Pino Ortega, J.; Silva, R.; Rico-González, M. Validity and reliability of inertial measurement units (IMUs) for jump height estimations: A systematic review. Human Movement 2021, 23, 1-20, doi:10.5114/hm.2023.111548.

also describe the IMU calibration procedures and how you check for the accuracy of these calibrations

Response: Thank you for your comment. We have explained the sensor calibration procedures, reinforcing them with a literature reference in the ‘Data Collection and Protocols’.

Please revise this phrasing for grammatical correctness.

Response: As suggested, we reviewed and corrected the sentence.

I cited 2 word tense errors....there appear to others - please find and correct them

Response: Thank you for noticing, we have changed it accordingly.

It seems to me that the narrative in 3.2 could be more concise, rather than repeat most values from Table 3; consider highlighting the most important results from Table 3 in the narrative

Response: We agree with the reviewer, but we only specified this component of the narrative so much because we wanted to comply with the rules for presenting the results of the t-test, including the value of t and the degrees of freedom. Nevertheless, following the reviewer's recommendation, we have simplified this narrative by highlighting only the main results and referring to Table 3.

Before restating your purpose briefly identify the problem you attempted to address and the rationale/need for this study.

Response: As suggested before recalling our objective, we briefly present the rationale for the study. In fact, this change helps the reader to get back in tune with the rationale of the study and thus to better prepare and take advantage of the discussion, thank you for the suggestion.

suggest: "The results of this study.....may be one of ..."I suggest "may be" because the results of one study are not definitive until it can be replicated

Response: Thanks for your comment. We understand your point of view, so we've adapted the sentence with ‘The results of this study... may be...’.

suggest: "This study's results..."

Response: Alteration done.

Round 2

Reviewer 1 Report

Comments and Suggestions for Authors

No further suggestions